

Experimental study of non-Darcian flow characteristics in permeable stones
Zhongxia Li[1], Junwei Wan[1], Tao Xiong[1], Hongbin Zhan[2]*, Linqing He[3], Kun Huang[1]*
[1]School of Environmental Studies, China University of Geosciences, 430074 Wuhan,
China.
[2]Department of Geology and Geophysics, Texas A & M University, College Station, TX
77843-3115, USA.
[3]Changjiang Institute of Survey Technical Research MWR, Wuhan, China.
* Corresponding authors:
Dr. Hongbin Zhan (zhan@geos.tamu.edu);
Dr. Kun Huang (cugdr_huang@cug.edu.cn).



**Abstract**
This study provides experimental evidence of Forchheimer flow and transition between
different flow regimes from the perspective of pore size of permeable stone. We have firstly
carried out the seepage experiments of permeable stones with four different mesh sizes,
including 24 mesh size, 46 mesh size, 60 mesh size, and 80 mesh size, which corresponding
to mean particle sizes (50% by weight) of 0.71 mm, 0.36 mm, 0.25 mm, and 0.18 mm. The
seepage experiments show that obvious deviation from Darcian flow regime is visible. In
addition, the critical specific discharge corresponding to the transition of flow regimes (from
pre-Darcian to post-Darcian) increases with the increase of particle sizes. When the "pseudo"
hydraulic conductivity ($K$) (which is computed by the ratio of specific discharge and the
hydraulic gradient) increases with the increase of specific discharge ($q$), the flow regime is
denoted as the pre-Darcian flow. After the specific discharge increases to a certain value, the
"pseudo" hydraulic conductivity begins to decrease, and this regime is called the post-
Darcian flow. In addition, we use the mercury injection experiment to measure the pore size
distribution of four permeable stones with different particle sizes, and the mercury injection
curve is divided into three stages. The beginning and end segments of the mercury injection
curve are very gentle with relatively small slopes, while the intermediate mercury injection
curve is steep, indicating that the pore size in permeable stones is relatively uniform. The
porosity decreases as the mean particle sizes increases, and the mean pore size can faithfully
reflect the influence of particle diameter, sorting degree and arrangement mode of porous
medium on seepage parameters. This study shows that the size of pores is an essential factor
for determining the flow regimes. In addition, the Forchheimer coefficients are also discussed
in which the coefficient $A$ (which is related to the linear term of the Forchheimer equation) is
linearly related to $1/d^2$ as $A = 0.0025\left(1/d^2\right) + 0.003$; while the coefficient $B$ (which is related
to the quadratic term of the Forchheimer equation) is a quadratic function of $1/d$ as





$B = 1.14\text{E-06}(1/d)^2 - 1.26\text{E-06}(1/d)$. The porosity ($n$) can be used to reveal the effect of
sorting degree and arrangement on seepage coefficient. The larger porosity leads to smaller
coefficients $A$ and $B$ under the condition of the same particle size.
***Keywords***: permeable stone, mercury injection experiment, pore size, flow regime, non-
Darcian flow.
**1. Introduction**
Darcy (1857) conducted a steady-state flow experiment in porous media and concluded
that the specific discharge was proportional to the hydraulic gradient, which is the Darcy's
law described as follow:

$$q = KJ \qquad (1\text{-}1)$$

where $q$ is the specific discharge, $J$ is the hydraulic gradient, and $K$ is the hydraulic
conductivity. However, when the specific discharge increases above a certain threshold,
deviation from Darcy's law is evident and the flow regime changes from Darcian flow regime
to the so called non-Darcian flow regime (Bear, 1972), which was first observed by
Forchheimer (1901), who proposed a widely used non-Darcian flow equation (the
Forchheimer equation) as follow:

$$J = Aq + Bq^2 \qquad (1\text{-}2)$$

where $A$ and $B$ are constants related to fluid properties and pore structure. The first and
second terms on the right side of Eq. (1-2) roughly reflect the contributions of viscous and
inertial forces (or resistance to flow), respectively.
From the Forchheimer equation, we can see that when the specific discharge is
sufficiently small, the inertial force can be ignored, and the equation is transformed to the





form of Darcy's law. On the other hand, when the specific discharge is sufficiently large, the
viscous force can be ignored, and the equation is transformed to the fully developed turbulent
flow.

In addition to the polynomial function such as the Forchheimer equation, there are also

several power-law functions proposed to describe the non-Darcian flow, and one of the most
commonly used power-law equations is the Izbash equation (Izbash, 1931), which is written
as:

$$J = aq^b \qquad (1\text{-}3)$$

where $a$ and $b$ are the empirical parameters that depend on flow and materials properties, and
the coefficient $b$ is usually between 1 and 2.

Because of its applicability for a wide range of velocity spectrum and its sound physics,

many scholars have adopted the Forchheimer equation (among many different types of
equations) to explore the non-Darcian flow. Besides, the theoretical background of the
Forchheimer equation has been discussed in details (Panfilov and Fourar, 2006). Numerous
experimental data have confirmed the validity of the Forchheimer equation for a variety of
nonlinear flow phenomena (Geertsma, 1974; Scheidegger, 1957; Wright, 1968). The
quadratic Forchheimer law has also been revealed as a result of numerical modelling by
simulating the Navier–Stokes flow in corrugated channels (Koch and Ladd, 1996; Skjetne et
al., 1999; Souto and Moyne, 1997). To sum up, the Forchheimer equation will be selected as
a representative to describe non-Darcy flow in this study.

Since the transition between Darcian flow and non-Darcian flow is important and

difficult to quantify, different scholars have carried out experiments using a wide range of
porous media, including homogeneous and heterogeneous porous media. Most of the
experimental studies have focused on the influence of mean particle size on flow state



transition using homogeneous porous media. In fact, it was believed that the nonlinear (or
non-Darcian) flow behavior in porous media was due to turbulent effect of flow in earlier
studies and the Reynold number ($Re$) was widely used to quantify the initiation of non-
Darcian flow. Bear (1972) concluded that the critical $Re$ (denoted as $Re_c$) of flow states (or
the $Re$ value at which flow starts to change from Darcian flow regime to non-Darcian flow
regime) is between 1 to 10. This finding was based on experimental data collected in packed
sand beds (Ergun, 1952; Fancher and Lewis, 1933; Lindquist, 1933; Scheidegger, 1960).
Schneebeli (1955) and Wright (1968) experimentally measured the value of $Re$ at the
beginning of turbulence and concluded that at very high velocities, the deviation from
Darcy's law is due to inertial effects followed by turbulent effects. In addition, Dudgeon
(1966) confirmed that $Re_c$ is about 60~150 for relatively coarse particle medium including
river gravels, crushed rock particles and glass marbles with grain sizes from 16 mm to 152
mm. Dudgeon (1966) indicated that the deviation from Darcy's law was not entirely due to
turbulence, but in a large extent due to inertial forces. Besides, Geertsma (1974) proposed an
empirical relationship among the inertial coefficient, permeability and porosity by conducting
non-Darcian flow experiments in unconsolidated and consolidated sands. The laser
anemometry and flow visualization studies of fluid flow in porous structures were used by
Dybbs and Edwards (1984), and they observed the nonlinear behavior at Reynolds numbers
around 150. Latifi et al. (1989) found that the transition from unsteady-state laminar flow to
non-Darcian flow in packed beds of spheres was between $Re$ values of 110 and 370. Seguin et
al. (1998) investigated the characterization of flow regimes in various porous media with
electrochemical techniques and found that the end of the Darcian flow regime in packed beds
of particles appeared at $Re$ about 180. Besides, Bu et al. (2014) indicated that the Darcian
flow in the packed beds would end at $Re$ around 100 by using electrochemical techniques.
Sedghi-Asl et al. (2014) found that the Darcy's law was usually not valid for rounded particle
sizes greater than 2.8 mm, according to the experimental results of flow in different sizes of
rounded aggregates. Our previous experimental research (Li et al., 2017) indicated that when
the particle size was smaller than 2.8 mm, the flow state gradually changed from pre-Darcy
flow to post-Darcy flow when the specific discharge increased. When the medium particle
sizes get even larger, such as 4.5 mm, 6.39 mm, 12.84 mm, and 16 mm (Moutsopoulos et al.,
2009), only the post-Darcy flow exists. Based on above analysis, we can see that many
previous experiments were carried out on homogeneous porous media, and the non-Darcy
flow characteristics are quite different in porous media with various particle sizes.
Among the numerous experimental studies on this issue, it is evident that most of them
focused on the effect of the mean particle size rather than the particle size distribution.
Recently, a few investigators recognized the importance of particle size heterogeneity in
understanding the transition of flow regimes, and have carried out a series of experiments to
address the issue. For instance, Van Lopik et al. (2017) provided new experimental data on
nonlinear flow behavior in various uniformly graded granular material for 20 samples,
ranging from medium sands ($d_{50}$ > 0.39 mm) to gravel ($d_{50}$ >6.34 mm). In addition, they
investigated the nonlinear flow behavior through packed beds of 5 different types of natural
sand and gravel from unconsolidated aquifers, as well as 13 different composite mixtures of
uniformly graded filter sands at different grain size distributions and porosity values (Van
Lopik et al., 2019). We have also discussed the effect of particle size distribution on
Forchheimer flow and transition of flow regimes in a previous study (Li et al., 2019). And our
study showed that the uniformity coefficient of porous media (a term used to describe the
pore size distribution) is a critical factor for determining the flow regimes besides the mean
particle sizes. Yang et al. (2019) investigated the effects of the particle size distribution on the
seepage behavior of a sand particle mixture subjected and evaluated the validity of empirical
formulas of permeability and inertia factor used in engineering practice. Shi et al. (2020)





discussed the non-Darcy flow behavior of granular limestone with a wide range of porosity
from 0.242 to 0.449. Based on the experimental data, Shi et al. (2020) proposed an empirical
hydraulic conductivity-porosity relation as well as an expression of inertial coefficient.
Regardless of the media investigated are homogeneous or heterogeneous, the essence of the
water passing capacity of porous media is pore sizes. Thus, exploring the distribution of pores
in porous media is the basis of studying flow dynamics of Darcian and non-Darcian flows.

The purpose of this study is to provide a quantitative analysis on the effects of pore size

on the transition of flow regimes between Darcian and non-Darcian flows based on a series of
laboratory experiments. To meet the objectives, we have firstly carried out the seepage
experiments of permeable stones with four different particle sizes. After that, we have
conducted mercury injection experiments on permeable stones with four different particle
sizes, and the pore size distributions with different particle sizes are obtained. Finally, the
effect of pore size on the transition of flow regimes and Forchheimer coefficients are
discussed based on the experimental results.
**2. Experimental methodology**
**2.1 Experimental setup and methods**

The experimental device is mainly composed of three parts: a water supply device, a

seepage experimental device and a measuring device. The schematic diagram of the
experimental apparatus is shown in Fig. 1. The water supply device consists of a tank, a
centrifugal pump and a flow regulating valve. The seepage experimental device consists of a
permeable stone and a plexiglass column. The measurement device monitors the real-time
water temperature and pressure. The water temperature is measured using a thermometer with
a precision of measurement of 0.1 °C. The water-level fluctuation is measured to calculate the
flow rate by a pressure transducer (CY201, Chengdu test LLC, China) in the range of 0–20
kPa with ±0.1% accuracy. The measuring device consists of a cylindrical tank and a pressure


transducer. The sample of permeable stone is 60 mm in length with a circular cross section of
51.3 mm in diameter. Two pressure transducers are set at the entrance and exit of the column
to measure the pressure drop. To minimize the boundary effects, the pressure transducer is
placed 30 mm away from either end of the column, and the way of pressure measurement is
consistent with our previous studies (Li et al., 2017; Li et al., 2019).

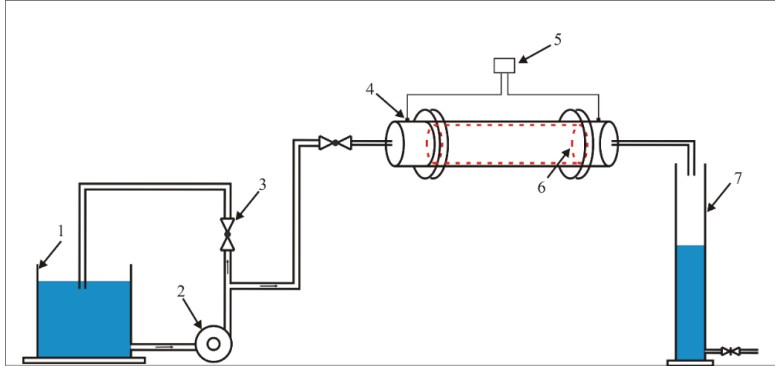


1) tank, 2) Pump, 3) Valve, 4) Pressure sensor, 5) Data collector, 6) Permeable stones, 7)
Measurement tank.

Fig. 1 The schematic diagram of experimental apparatus.

**2.2 Experimental Materials and Procedures**

Four different particle sizes of permeable stones are selected to carry out the seepage

experiment in this study. It is necessary to make a brief overview of the preparation process
of permeable stone, which is a type of artificially made tight porous medium formed by sand
grains and cementing compound. In the process of preparing permeable stone, a certain
particle size of sand and cementing compound is put in a mold, and is consolidated at room
temperature. We have carried out the seepage experiments of permeable stones with four
different mesh sizes, including 24 mesh size, 46 mesh size, 60 mesh size, and 80 mesh size,
and the mesh size is defined as the number of mesh elements (all in square shapes) in a one
inch by one inch square, thus a greater number of mesh size implies a smaller particle size.





For instance, we can convert above four different mesh sizes of permeable stones into
corresponding particle sizes of 0.71 mm, 0.36 mm, 0.25 mm and 0.18 mm, respectively. The
pore structure of permeable rock will not change in the process of the seepage experiment
under room temperature, and the physical diagrams of four kinds of permeable stones with
different particle sizes are shown in Fig. 2 and Fig. 3.

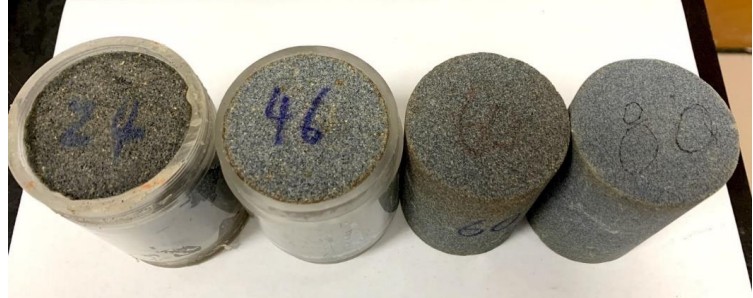


Fig. 2 Physical drawing of permeable stones with four different particle sizes.

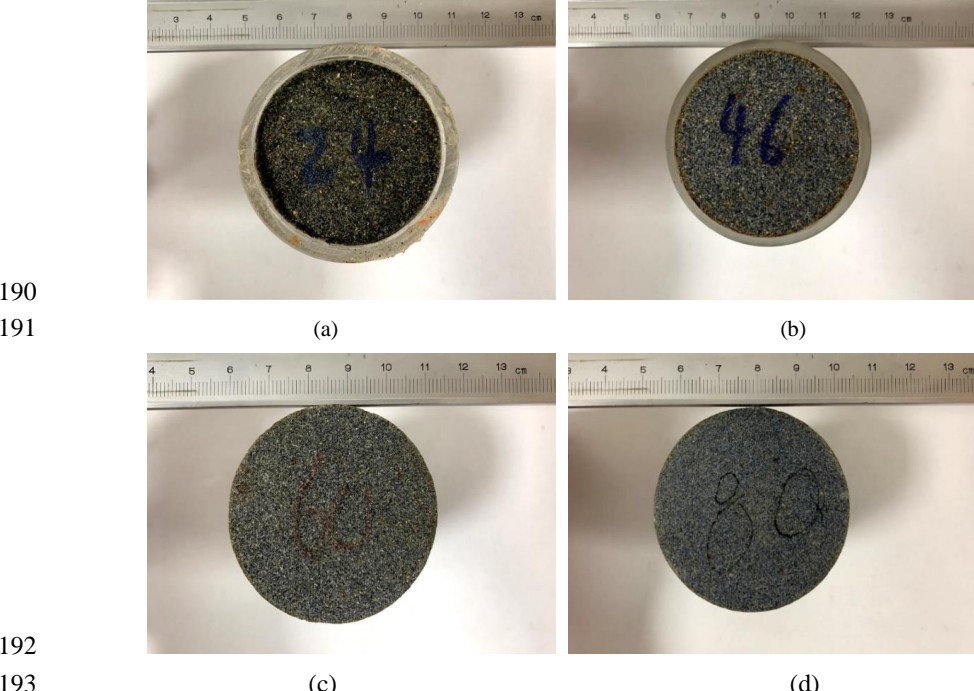

(a)                                                    (b)

(c)                                                    (d)

Fig. 3 Permeable stones with different particle sizes: (a) 24 mesh size or 0.71 mm, (b) 46

mesh size or 0.36 mm, (c) 60 mesh size or 0.25 mm, and (d) 80 mesh size or 0.18 mm.





It is worth mentioning that the contact surface of the sample and the plexiglass column
is sealed to prevent any preferential flow through the wall of the plexiglass column. After the
permeable stone is inserted into the plexiglass column, both ends are sealed with silicone glue.
The water passing through the permeable stone is then collected by a cylindrical tank.
Moreover, the ratio of the internal diameter of the column to the particle size of permeable
stone is greater than 12, which can eliminate any possible wall effect on the seepage
according to Beavers et al. (1972). When carrying out the experiment, it usually takes about
two hours to saturate the permeable stone. For each packed sample, more than 25 tests with
different constant inlet pressures were conducted under steady-state flow condition. In
addition, for each group of permeable stone, repeated tests under the same experimental
condition were carried out 3-4 times to ensure the accuracy of the results.
**3. Results and discussion**
**3.1 Permeable stone seepage experiment**
In this study, we selected permeable stone with four different particle sizes as the
research objects, including 24 mesh size, 46 mesh size, 60 mesh size and 80 mesh size. The
mesh size is the number of holes per inch of screen mesh and the particle size is inversely
proportional to the mesh size. The mean particle sizes corresponding to the four different
mesh sizes are 0.71 mm, 0.36 mm, 0.25 mm, and 0.18 mm, respectively, where the mean
particle size is corresponding to 50% by weight hereinafter in this study. Such a definition of
mean particle size may be different from some other studies such as Fetter (2001) which has
used 10% by weight as the mean particle size. The relationship between the specific
discharge ($q$) and the hydraulic gradient ($J$) of permeable stones is plotted in Fig. 4. The units
of specific discharge mentioned in this study are all converted to meters per day (m/d).



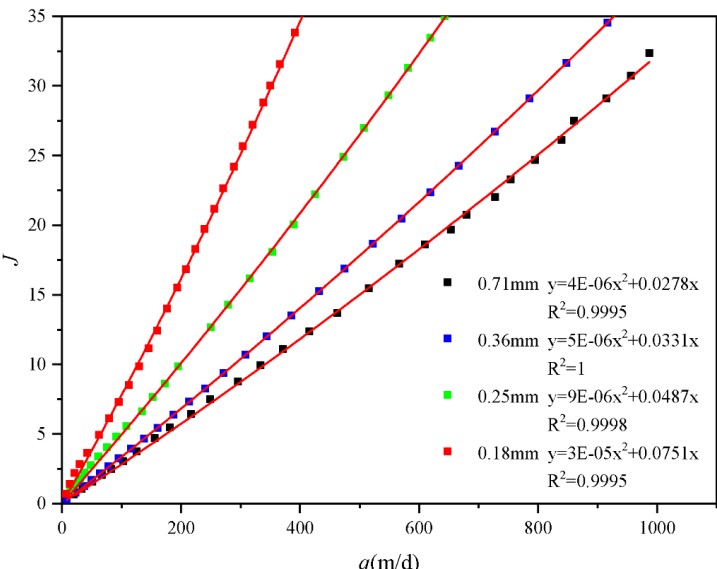


Fig. 4 Variation of $J$ with $q$ of four permeable stones with different particle sizes.


Fig. 4 shows that when $q$ is somewhat the same, a larger mesh size (which means a


smaller particle size) will lead to a larger $J$. And the results are consistent with our previous


studies (Huang et al., 2013; Li et al., 2017; Li et al., 2019). However, the nonlinear


characteristics of $q$-$J$ curve are not obvious due to the relatively small velocity range used in


the experiments. Nevertheless, the best-fitting results using the Forchheimer equation are


satisfactory. To analyze the influence of pore size on seepage flow regimes, we have obtained


the relationship between $q$ and the "pseudo" hydraulic conductivity ($K$) (which is computed


using $q/J$) of four permeable stones with different particle sizes, as shown in Fig. 5. We


should point out that the "pseudo" hydraulic conductivity term discussed here for non-


Darcian flow is usually not a constant, thus it is different from the hydraulic conductivity


term used in Darcy's law, which is a constant. It is obvious that the hydraulic conductivity is


not a constant with the increase of specific discharge, so it is called the "pseudo" hydraulic


conductivity (Li et al., 2019).




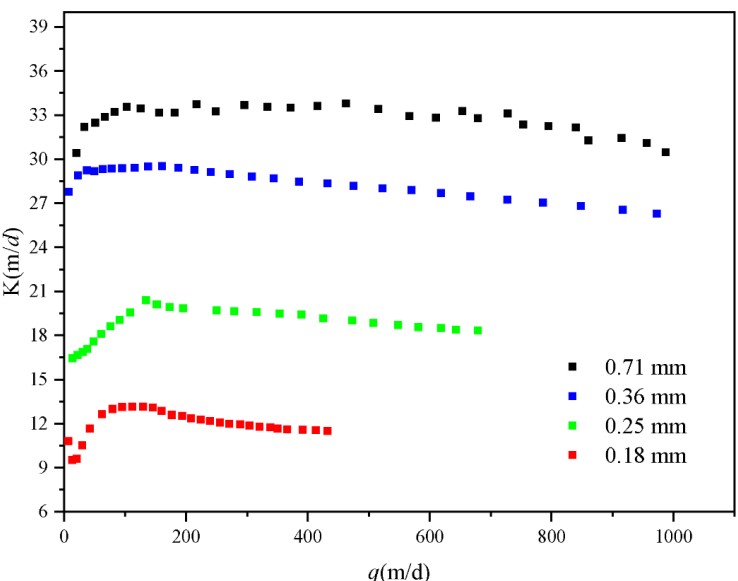


Fig. 5 Variation of $K$ with $q$ of four permeable stones with different particle sizes.

We can divide the $q$-$K$ curve into two segments: for the first segment, $K$ increases with

the increase of $q$, which is denoted as the pre-Darcian flow. For the second segment, after $q$
increases to a certain value, $K$ begins to decrease with $q$, which is called the post-Darcian
flow. When the hydraulic gradient is small (and $q$ is small as well), a great portion of water is
bounded (or becomes immobile) on the surface of solids due to the solid-liquid interfacial
force, and only a small fraction of the water is mobile and free to flow through the pores. As
the hydraulic gradient increases (and $q$ increases as well), the initial threshold for mobilizing
the previously immobile water near the solid-liquid surface is overcome and more water
participates in the flow. For this reason, the "pseudo" hydraulic conductivity increases with
the increase of hydraulic gradient and the specific discharge in the first segment. When the
specific discharge increases to the critical specific discharge ($q_c$), the "pseudo" hydraulic
conductivity is maximized. According to $K = \dfrac{q}{Aq + Bq^2} = \dfrac{1}{A + Bq}$ based on Eq. (1-2), we can
find that the "pseudo" hydraulic conductivity begins to decrease as the specific discharge





continues to increase. Besides, the critical specific discharge corresponding to the transition
of flow regimes (from pre-Darcian to post-Darcian) increases with the increase of particle
sizes (or decrease of mesh sizes).

**3.2 Mercury injection experiment**

The particle size, different grain size distributions and degree of sorting are the main
factors that determine the size and shape of pores. And the shape of the pores determines the
tortuosity and distribution of flow paths, which are related to viscous and inertial flow
resistances. It is generally accepted in previous studies that the pore sizes of porous media
have an impact on the seepage law (Maalal et al., 2021; Zhou et al., 2019). However, the
structure of natural porous media is very complex, and it is difficult to quantify the effects of
the arrangement of particles on the seepage law. The characteristics of pore size distribution
contains critical information for quantifying the flow regimes. The mercury intrusion
porosimetry and the nitrogen adsorption isotherm are two commonly used methods to
characterize the pore sizes and their distribution (Rijfkogel et al., 2019). Besides, other
techniques can also be used to derive the pore size distribution, such as small-angle neutron
and X-ray scattering measurements, CT images and nuclear magnetic resonance (Anovitz and
Cole, 2015; Hall et al., 1986; Kate and Gokhale, 2006; Lindquist et al., 2000). In this study
we will use the mercury injection experiment to measure the pore size distribution of the four
permeable stones with different particle sizes and use the information to describe the flow
regimes.
To quantitatively study the pore size and pore throat distribution, we need to envisage a
physically based conceptual model to describe the pore structures of permeable stones. The
commonly used model is the so-called capillary model (Pittman, 1992; Rezaee et al., 2012;
Schmitt et al., 2013), which approximates the connected pores as many paralleled capillaries.
And the capillary forces are generated at the phase interface due to the surface tension



between the solid and liquid phases when liquid flows in a capillary. The capillary force is
directed toward the concave liquid level, and is shown as (Washburn, 1921):

$$P_c = \frac{2\sigma\cos\theta}{r}$$    (3-1)

where $P_c$ is the capillary force, $\sigma$ is the solid-liquid interfacial tension, $\theta$ is the wet angle
between the liquid and the solid surface, and $r$ is the radius of curvature in capillary.

Since mercury is a nonwetting phase to solids, so to get mercury into the pores of the

permeable stone, an external force (or displacement pressure) must be applied to overcome
the capillary force. When a greater pressure is applied, mercury can enter smaller pores.
When a certain pressure is applied, the injection pressure is equivalent to the capillary
pressure in the corresponding pore. Then we can calculate the corresponding capillary radius
according to Eq. (3-1), and the volume of mercury injected is the pore volume.

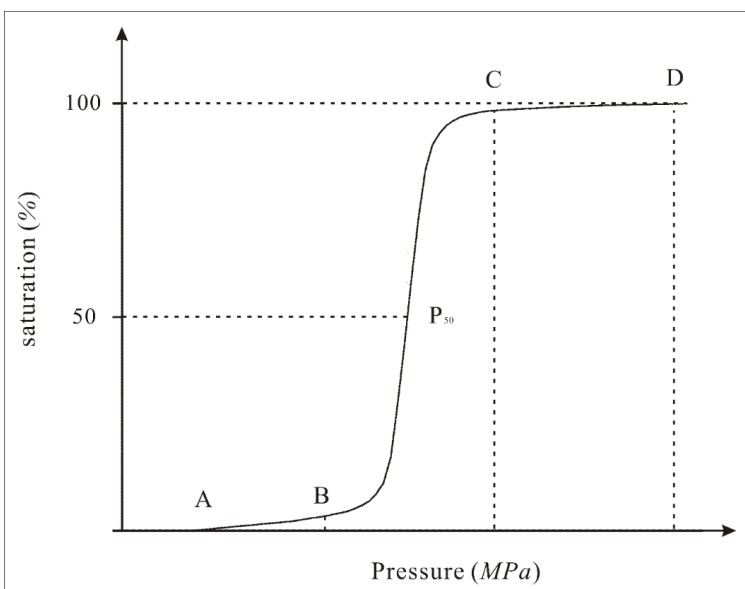


Fig. 6 Schematic diagram of pressure changes with saturation: the initial stage (A-B), the

intermediate mercury entry stage (B-C), and the end stage (C-D).





By continuously increasing the injection pressure, one can obtain the curve of injection
pressure and the volume of injected mercury, from which one can also obtain the pore-throat
distribution curve and capillary pressure curve. According to the amount of mercury injected
at different injection pressures, the relation between the injection pressure and the injection
saturation is shown in Fig. 6.
Fig. 6 shows that the mercury injection curve can be divided into three stages. Firstly,
during the initial stage (A-B) which has a very mild slope, the intake pressure is very small
and the intake saturation is also very low. With the increasing of the injection pressure, the
intake saturation slowly increases. Secondly, during the intermediate mercury entry stage (B-
C) which has a steep slope, a small pressure change will lead to a significant saturation
change. This means that the pores are relatively uniform and the differences in pore sizes are
small. Hence, we can use the pressure ratios of B and C ($P_C/P_B$) to reflect the inhomogeneity
of the pore size in the porous media. Besides, when the saturation reaches 50%, the
corresponding pressure value ($P_{50}$) reflects the characteristics of the mean pore size, and a
larger $P_{50}$ leads to a larger mean pore size. Finally, during the end stage (C-D) which has a
very mild slope as well, the amount of mercury will not increase considerably when the
injection pressure increases. This indicates that nearly all the pores are essentially filled with
mercury, and the mercury injection experiment is completed. After completing the mercury
injection experiments, we have obtained the mercury injection curves of four permeable
stones with different particle sizes, as shown in Fig. 7.
We can make a number of interesting observations based on Fig. 7. Firstly, the pressure
at the starting point (when the saturation begins to increase), denoted as $P_A$, increases as the
mean particle size decreases. This means that the maximum pore size in permeable stone
decreases with the decrease of the mean particle sizes. Secondly, the mercury injection curves
of four permeable stones all include steep intermediate stages, indicating that the pore size





distributions are all relatively uniform. And the corresponding pressure values at points B and
C increase as the mean particle sizes decreases. Moreover, the pressure ratios corresponding
to points B and C ($P_C/P_B$) also decrease with the decrease of particle sizes, suggesting even
more uniform pore size distributions with decreasing particle sizes. Thirdly, the intermediate
mercury entry stages gradually shift to the right with the decrease of particle sizes. When the
saturation reaches 50%, the corresponding pressure (the median pressure) decreases with the
increase of the mean particle sizes. Fourthly, the mercury injection curves of these four
permeable stones with different particle sizes all approach 100% saturation with very mild
slopes, indicating that there are few small pores in the permeable stones. We have extracted
the key pressure characteristic values of mercury injection experiment of Fig. 7, and listed the
results in Table 1.

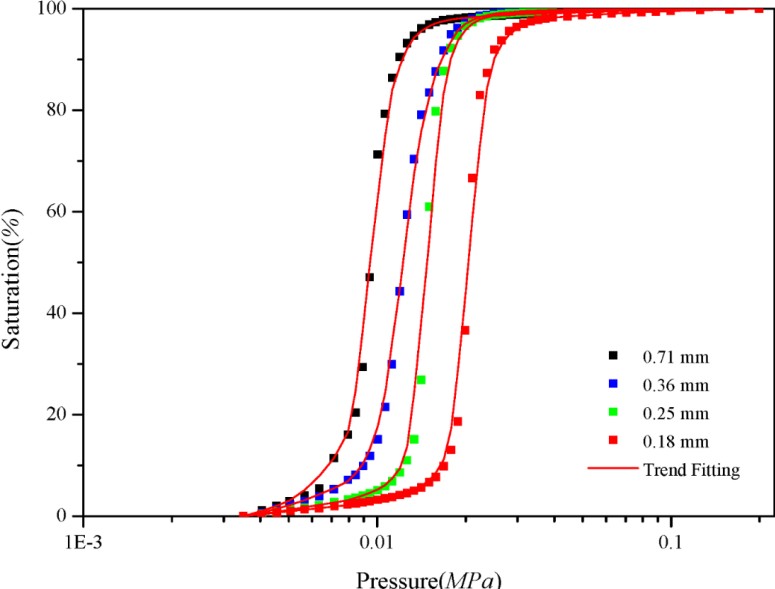


Fig. 7 Variation of pressure with saturation of four permeable stones with different particle
sizes.





Table 1. Pressure characteristic values of four permeable stones with different particle sizes.

| Mesh size | $P_A(MPa)$ | $P_B(MPa)$ | $P_C(MPa)$ | $P_{50}(MPa)$ | $P_C/P_B$ |
|---|---|---|---|---|---|
| 24 | 0.0041 | 0.0064 | 0.0133 | 0.0094 | 2.0987 |
| 46 | 0.0045 | 0.0071 | 0.0188 | 0.0119 | 2.6374 |
| 60 | 0.0051 | 0.0112 | 0.0211 | 0.0150 | 1.8764 |
| 80 | 0.0057 | 0.0158 | 0.0281 | 0.0211 | 1.7758 |

To observe the pore size distributions of the four permeable stones with different particle
sizes in more details, we can calculate the percentages of different pore sizes in permeable
stones according to the mercury injection curves, as shown in Figs. 8-11.

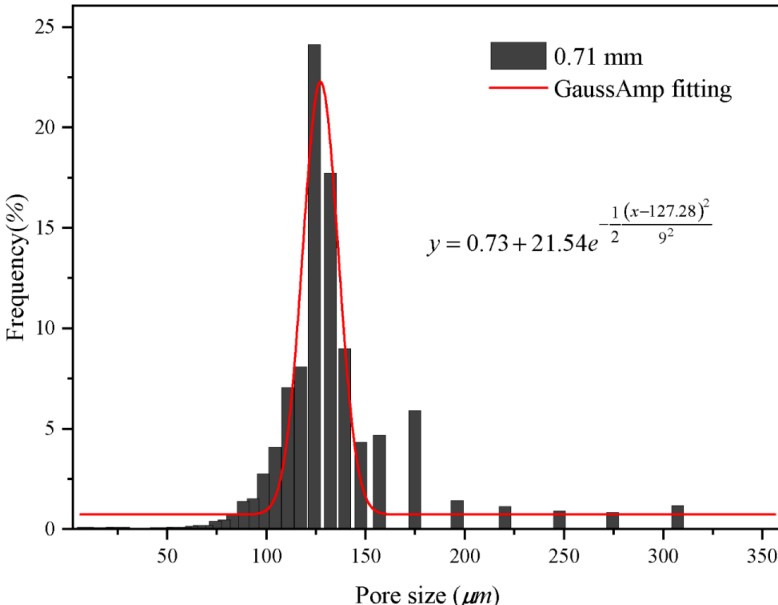


Fig. 8 Histogram of pore size distribution of permeable stone with diameter of 0.71 mm.





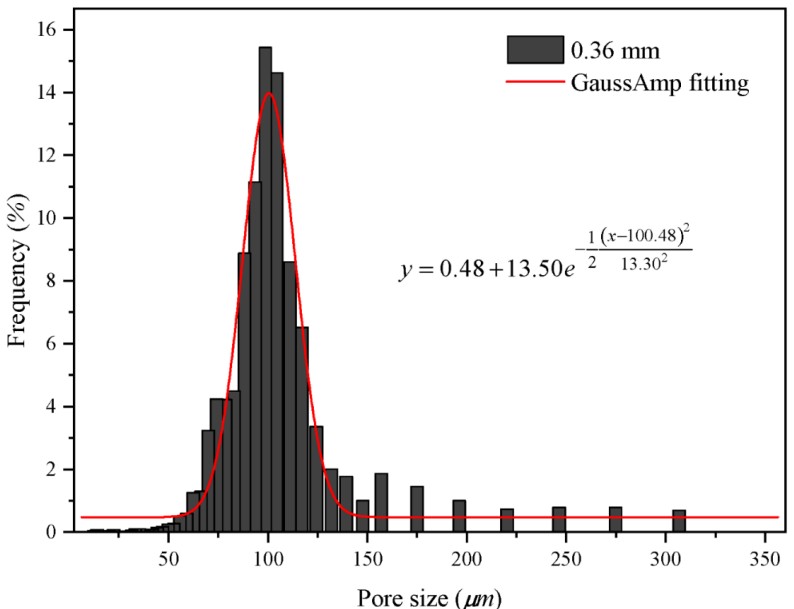


Fig. 9 Histogram of pore size distribution of permeable stone with diameter of 0.36 mm.

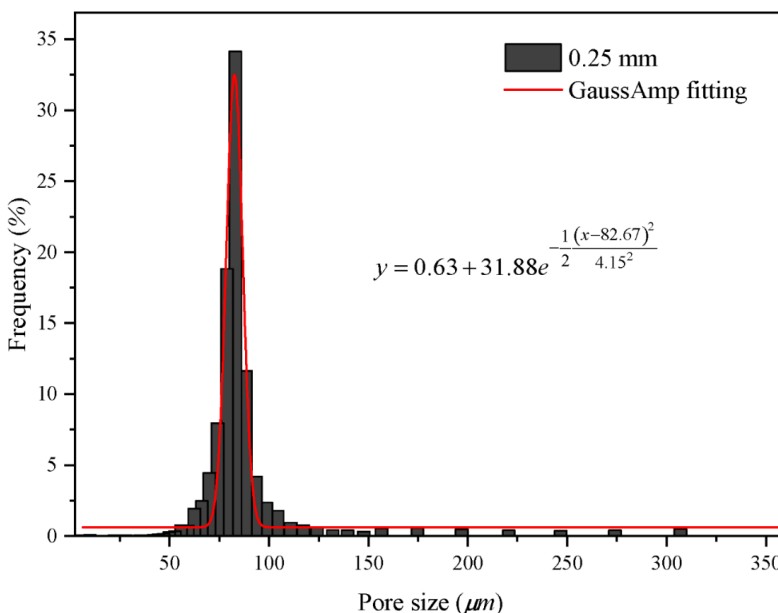


Fig. 10 Histogram of pore size distribution of permeable stone with diameter of 0.25 mm.




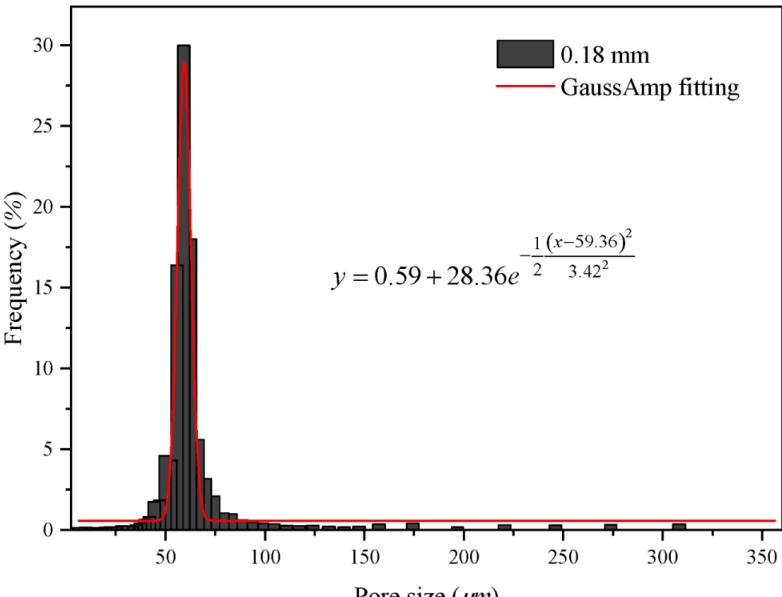


Fig. 11 Histogram of pore size distribution of permeable stone with diameter of 0.18 mm.
From Fig. 8 to Fig. 11 we can find that the pore sizes of the four permeable stones are
uniform and fall within narrow ranges. The pore size distributions of four different particle
sizes show a skewed normal distribution. Besides, the pore maximum proportion (the peak of
the curve, see Figs. 8-11) of permeable stones with different particle sizes are different, which
are 124 $\mu m$, 99 $\mu m$, 83 $\mu m$ and 59 $\mu m$, respectively. The Gaussian function is widely used to
characterize the pore system and classify the petrophysical rock (Harlan et al., 1995; Jeon et
al., 2014; Xu and Torres-Verdín, 2013), and the general form of the Gauss function is shown
below:

$$y = y_0 + He^{-\frac{(x-x_c)^2}{2w^2}} \qquad (3\text{-}2)$$

where $H$ is the height of the peak of the mercury injection curve, $x_c$ is the abscissa
corresponding to the peak of the curve (the pore size), $w$ is the standard variance, which
represents the width of the curve. To characterize the distribution of pore structure of four





different permeable stones, we best-fit the Gaussian curve of the pore distribution of four
permeable stones with different particle sizes, and the best-fitted parameters are shown in
Table 2. We can make several interesting observations from Table 2. Firstly, the expected
value ($x_c$) decreases with decreasing particle sizes of permeable stone, and the $x_c$ values of
different permeable stones are almost the same. Secondly, the standard variance ($w$)
corresponding to the permeable stone of 0.18 mm is the smallest, indicating that the pore size
distribution is more concentrated (or relatively homogeneous). For comparison, the pore size
distribution of 0.36 mm permeable stone is the widest with the greatest variance. Finally,
different values of $H$ represent different proportions of pore sizes, among which the highest
proportion can reach 34.04%. It will be desirable to establish a correlation between the
parameters used in the pore-size distribution of Eq. (3-2) with the two Forchheimer
coefficients $A$ and $B$. This objective may be achieved using high-resolution pore-scale fluid
mechanics simulations, which are out of the scope of this study. Further research is needed to
address this issue in the future.
Table 2. Gaussian function characteristic values of four permeable stones with different
particle sizes.

| Mesh size | particle size (mm) | $y_0$ | $H$ | $x_c$ | $w$ |
|-----------|--------------------|-------|-----|-------|-----|
| 24 | 0.71 | 0.73 | 21.54 | 127.28 | 9.00 |
| 46 | 0.36 | 0.48 | 13.49 | 100.48 | 13.30 |
| 60 | 0.25 | 0.63 | 31.88 | 82.67 | 4.15 |
| 80 | 0.18 | 0.59 | 28.36 | 59.36 | 3.42 |






The pore size distributions fall within ever narrower ranges with mesh sizes become

larger. Moreover, the cumulative percentage frequency curves of the pore size distributions

with different particle sizes are exhibited in Fig. 12 and the results are shown in Table 3.

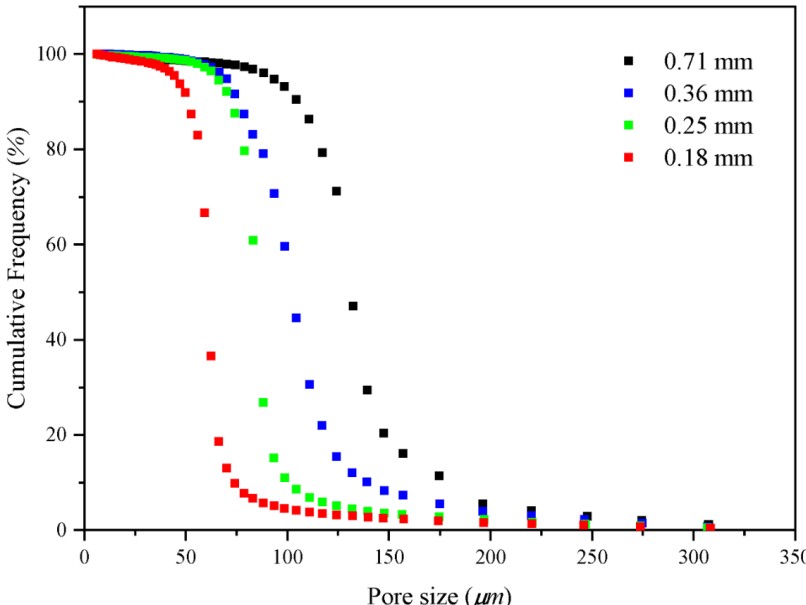


Fig. 12. The cumulative frequency curve of pore size distribution.

Fig. 12 shows that $R_{50}$ (the pore size corresponding to the median pressure $P_{50}$) increases

with the increase of permeable stone particle size, and the mean pore diameter ($R_m$) also

increases. In general, the pore size corresponding to the median pressure (denoted as $R_{50}$)

may be slightly different than the mean pore diameter ($R_m$) which has been defined in

different ways by various investigators when analyzing the pore size distributions (Hea and

Zhangb, 2015; Zhen-Hua et al., 2007; Zhihong et al., 2000). As $R_{50}$ is easily identifiable in

the mercury injection experiments, it is used in this study as a representative of the mean pore

diameter ($R_m$) of the permeable stone. Besides, the seepage law of permeable stone is closely

related to the pore size, and the smaller average pore size will result in a larger hydraulic

gradient under the condition of the same specific discharge (see Fig. 4). The pore size





characteristic values with different particle sizes are listed in Table 3. We find that the
porosity decreases as the particle size increases while the mean pore diameter increases. And
the mean pore size can reflect the influence of particle diameter, sorting degree and
arrangement mode of porous medium on seepage parameters.
Table 3. Pore size characteristic values of four permeable stones with different particle sizes.

| Mesh size | Mean particle size ($mm$) | Porosity (%) | $R_m$ ($\mu m$) | $R_{50}$ ($\mu m$) |
|---|---|---|---|---|
| 24 | 0.71 | 32.35 | 131.31 | 131.34 |
| 46 | 0.36 | 36.69 | 102.56 | 103.42 |
| 60 | 0.25 | 40.82 | 84.73 | 85.09 |
| 80 | 0.18 | 42.88 | 60.97 | 61.12 |

*Note: $R_m$ is the mean pore diameter, $R_{50}$ is the pore diameter corresponding to the median
pressure $P_{50}$.

### 392    3.3 Analysis of influencing factors of Forchheimer equation coefficients

### 393    3.3.1 Influence of particle size on equation coefficient

The analysis of non-Darcy coefficient has always been of interest to many researchers

working in different disciplines of porous media flow (Moutsopoulos et al., 2009; Sedghi-Asl
et al., 2014; Shi et al., 2020). Various studies have suggested expressions for Forchheimer
coefficients, Ward (1964) proposed the estimation formula of Forchheimer coefficients $A$ and
$B$ by analyzing the experimental data of 20 different porous media:

$$A = \frac{360}{gd^2} \qquad B = \frac{10.44}{gd} \qquad\qquad (3\text{-}3)$$





where $d$ is the particle diameter hereinafter, $g$ is the acceleration due to gravity. Based on the
mixed model of parallel capillary tubes, Blick (1966) proposed a new form of Forchheimer
coefficients:

$$A = \frac{32}{gnd^2} \qquad B = \frac{C_D}{2gn^2d} \tag{3-4}$$

where $n$ is the porosity of the medium, $C_D$ is an appropriate phenomenological coefficient.
Besides, Ergun (1952) suggested the new expressions by extending the classic Kozeny-
Carman model:

$$A = \frac{150(1-n)^2}{gn^3d^2} \qquad B = \frac{1.75(1-n)}{gn^3d} \tag{3-5}$$

Kovács (1981) analyzed 300 data in the range of $10 < Re < 100$, and derived a similar
expression for the case of dispersed spherical particles, and the coefficients of Forchheimer
are as follows:

$$A = \frac{144(1-n)^2}{gn^3d^2} \qquad B = \frac{2.4(1-n)}{gn^3d} \tag{3-6}$$

Kadlec and Knight (1996) also proposed the following Forchheimer coefficients:

$$A = \frac{255(1-n)^2}{gn^{3.7}d^2} \qquad B = \frac{2(1-n)}{gn^3d} \tag{3-7}$$

Sidiropoulou et al. (2007) focused on the determination of the Forchheimer coefficients
for non-Darcian flow in porous media and evaluated the original theoretical equations above
and the validity of these equations was checked using existing experimental data. In addition,
the Root Mean Square Error (RMSE) was used as a criterion to quantitatively evaluate the





coefficients, and the RMSE was defined as $RMES = \sqrt{\dfrac{\sum_{i=1}^{N}(x_i - y_i)^2}{N}}$ , where $x_i$ were the
experimental values of Forchheimer coefficients, $y_i$ were the values computed by different
equations above, and $N$ was the total number of experimental points (Moutsopoulos et al.,
2009). The different forms of Forchheimer coefficients described above are based on different
assumptions and simplifications of pore structure. Consequently, these series of coefficients
are applicable under specific conditions with different degrees of accuracy.

According to Eq. (1-2), the hydraulic gradient ($J$) is composed of a viscous force-related

component ($J_n$) and an inertia force-related component ($J_r$), as below:

$$J_n = Aq = \frac{\alpha\mu}{\rho g}\frac{1}{d^2}q \quad J_r = \frac{\beta}{g}\frac{1}{d}q^2 \qquad (3\text{-}8)$$

We can see from Eq. (3-8) that the $J_n$ is inversely proportional to the square of the particle
size, and the $J_r$ is inversely proportional to the particle size when the specific discharge
remains the same. Both $J_n$ and $J_r$ are closely related to specific surface area and sizes of pores.
Therefore, the particle size is an important factor affecting the Forchheimer coefficient,
Huang et al. (2013) carried out the experimental investigation on water flow in four columns
with cubic arrays of acrylic balls in diameter 3 mm, 5 mm, 8 mm and 10 mm, where all the
acrylic balls are arranged in regular cubes. Accordingly, the coefficients $A$ and $B$ can be
written as follows:

$$A = \frac{\alpha\mu}{\rho g}\frac{1}{d^2} \qquad B = \frac{\beta}{g}\frac{1}{d} \qquad (3\text{-}9)$$

where $\alpha$ and $\beta$ are constants related to the shape, sorting, and arrangement of the particles,
and the specific derivation process is detailed in the previous study (Huang, 2012). The
experimental results showed that the coefficient $A$ was inversely proportional to the particle
diameter square ($d^2$) and coefficient $B$ was inversely proportional to the particle size ($d$)





(Huang et al., 2013).
The uniform diameter cubic arrangement of porous media mentioned above is a rather
ideal medium. The shape and arrangement of particles of natural pore aquifers are usually
irregular. Therefore, the above-mentioned linear correlations between $A$ and $1/d^2$, and
between $B$ and $1/d$ should be examined specifically. For this purpose, we collect the
experimental data of homogeneous porous media, including the previous research results and
the results of other scholars. Among them, samples P1-P4 are the permeable stones selected
in this study, samples L1-L5 are from previous studies (Li et al., 2017), and the experimental
data of samples M1-M4 are from Moutsopoulos et al. (2009). The fitting coefficients are
shown in Table 4. Furthermore, we can identify nice correlations between the Forchheimer
coefficient $A$ and $1/d^2$ and between the Forchheimer coefficient $B$ and $1/d$, which are shown
in Fig. 13 and Fig. 14, respectively.

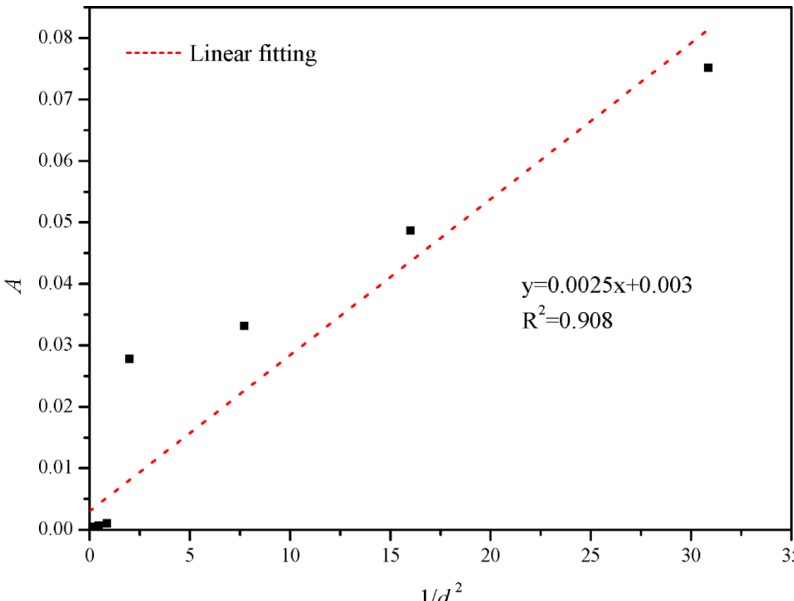


Fig. 13. Variation of $A$ with $1/d^2$ of different homogeneous particle sizes.





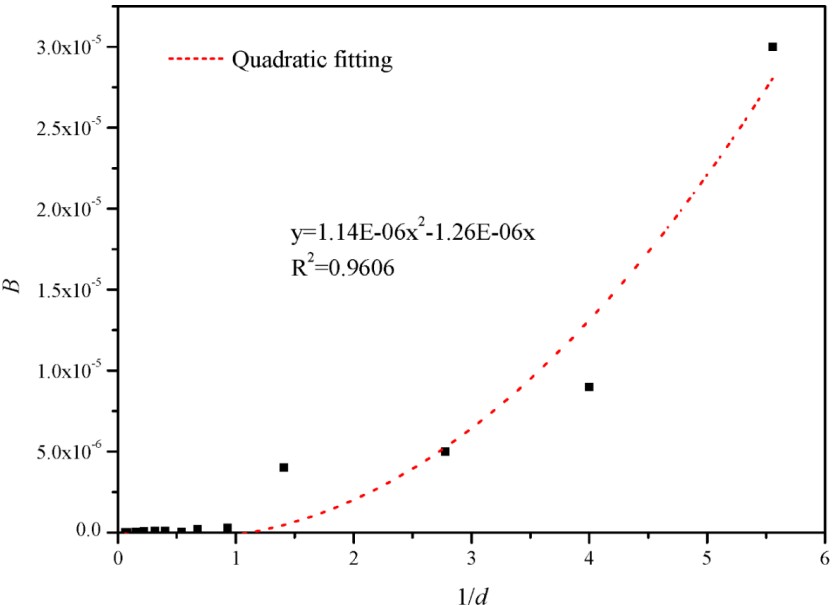


Fig. 14. Variation of $B$ with $1/d$ of different homogeneous particle sizes.

We can see from Fig. 13 that the coefficient $A$ is linearly related to $1/d^2$ and the

relationship between coefficient $A$ and is given as $A = 0.0025\left(1/d^2\right)+0.003$. And the
relationship between coefficient $B$ and $1/d$ is completely different from the linear correlation
as reported before. Fig. 14 shows that the coefficient $B$ is quadratic related to $1/d$ and the
relationship between coefficient $B$ and $1/d$ is given as $B = 1.14\text{E-06}\left(1/d\right)^2 - 1.26\text{E-06}\left(1/d\right)$.
That is to say, the relationship between coefficient $A$ and $1/d^2$ is consistent with the law of
simple cubic arrangement porous media, but the relationship between coefficient $B$ and $1/d$ is
not consistent with the law of simple cubic arrangement porous media. The structure of
porous medium arranged in cubes is different from the permeable stone. The porosity of the
porous media with spheres arranged in cubic is close to 0.48, independent of the diameter of
spheres. While the particle shape, arrangement and tightness of permeable stone are different,
and the porosity of permeable stone with different particle size is also different (see Table 3).





Table 4. Experimental fitting coefficient of different homogeneous particle sizes.

| Sample | Particle size (mm) | Fitting equation | A | B | The correlation |
|--------|--------------------|------------------|---|---|------------------|
| P1 | 0.18 | $y=0.0751x+3E\text{-}05x^2$ | 0.0751 | 3E-05 | 0.9995 |
| P2 | 0.25 | $y=0.0487x+9E\text{-}06x^2$ | 0.0487 | 9E-06 | 0.9998 |
| P3 | 0.36 | $y=0.0331x+5E\text{-}06x^2$ | 0.0331 | 5E-06 | 1 |
| P4 | 0.71 | $y=0.0278x+4E\text{-}06x^2$ | 0.0278 | 4E-06 | 0.9995 |
| L1 | 1.075 | $y=0.001x+3E\text{-}07x^2$ | 0.001 | 3E-07 | 0.9999 |
| L2 | 1.475 | $y=0.0007x+2E\text{-}07x^2$ | 0.0007 | 2E-07 | 0.9998 |
| L3 | 1.85 | $y=0.0005x+5E\text{-}08x^2$ | 0.0005 | 5E-08 | 0.9998 |
| L4 | 2.5 | $y=0.0005x+9E\text{-}08x^2$ | 0.0005 | 9E-08 | 0.9997 |
| L5 | 3.17 | $y=0.0004x+1E\text{-}07x^2$ | 0.0004 | 1E-07 | 0.9998 |
| M1 | 4.5 | $y=3E\text{-}05x+7E\text{-}08x^2$ | 3E-05 | 7E-08 | 0.9913 |
| M2 | 6.39 | $y=3E\text{-}05x+3E\text{-}08x^2$ | 3E-05 | 3E-08 | 0.9984 |
| M3 | 12.84 | $y=1E\text{-}05x+2E\text{-}08x^2$ | 1E-05 | 2E-08 | 0.9977 |
| M4 | 16 | $y=1E\text{-}05x+2E\text{-}08x^2$ | 1E-05 | 2E-08 | 0.998 |

**3.3.2 Influence of porosity on equation coefficient**

In above sections, we have analyzed the influence of particle sizes on seepage

coefficient. Furthermore, the pore size and pore specific surface area are also related to the
arrangement and sorting degree of particles, that is, to the porosity of porous media. To
explore the effect of sorting degree on seepage coefficient, we draw a schematic diagram of



different sorting degree of particles, as shown in Fig. 15. The degree of particle sorting is one
of the important factors affecting the pore size. In porous media with a poor sorting degree,
the pore size is usually determined by the diameter of the smallest particle. We can see from
Fig. 15 that the pores between the larger particles are filled by smaller particles, resulting in
even smaller pores. In addition, the poorer sorting degree of particles leads to the larger pore
specific surface area and stronger viscous force of flow, which can lead to a larger coefficient
$A$.

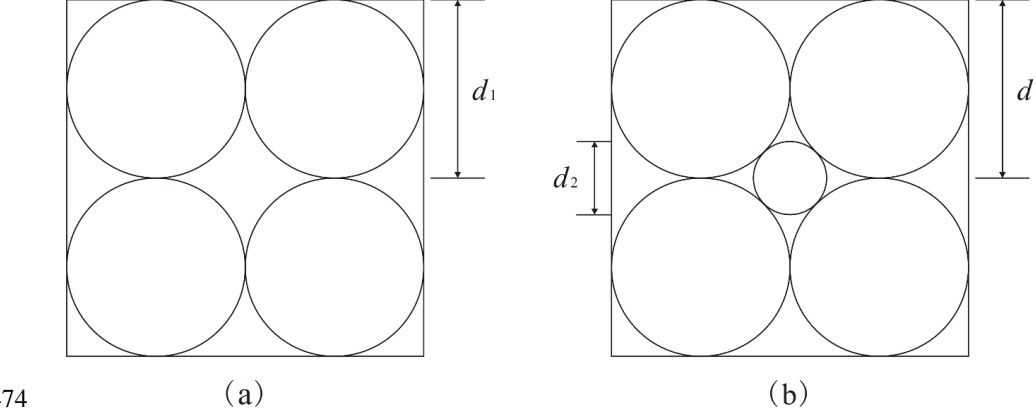

(a)                                (b)


Fig. 15. The schematic diagram of particle sorting degree in different types.

Furthermore, we have also provided the schematic diagrams of spherical particles with

equal size in two simple arrangements, namely cube arrangement and hexahedron
arrangement, as shown in Fig. 16. And the cube arrangement is the less compact arrangement
with a pore diameter of $0.414d$, while the hexahedron arrangement is the more compact
arrangement with a pore diameter of $0.155d$. The characteristic value of pore structure in
different arrangement with the same particle size are shown in Table 5. We can see that
different arrangement modes will substantially affect the pore specific surface area and pore
size of porous media. The more compactly packed particles lead to the larger pore specific
surface area and stronger viscous force. Meanwhile, the smaller pore diameter is associated




with stronger effect of viscous force and inertia force. In summary, the better sorting degree
of particles leads to the weaker viscous and inertial forces, then the coefficients *A* and *B* will
be smaller. As the better sorting degree and the less compact (or looser) arrangement particles
mean the larger porosity, so we can conclude that the larger porosity leads to the smaller
coefficients *A* and *B* under the condition of the same particle size.

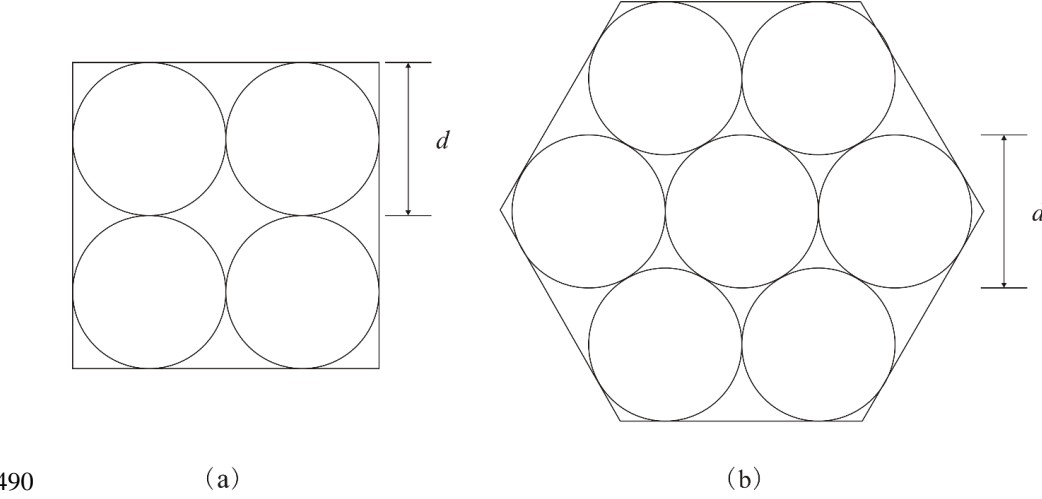

(a)               (b)

Fig. 16. The schematic diagram of particle arrangement in different types.

Table 5. Characteristic value of pore structure in different arrangement with the same particle
size.

| Arrangement mode | Side length | Porosity (%) | Specific surface area |
| --- | --- | --- | --- |
| Cube | $2d$ | 47.60 | 3.142 |
| Hexahedron | $1.577d$ | 43.30 | 3.402 |

However, the structure of natural porous media is much more complex and

heterogeneous than what has been shown in Figure 16, so it is difficult to quantitatively
describe the effect of sorting degree and arrangement on seepage law.





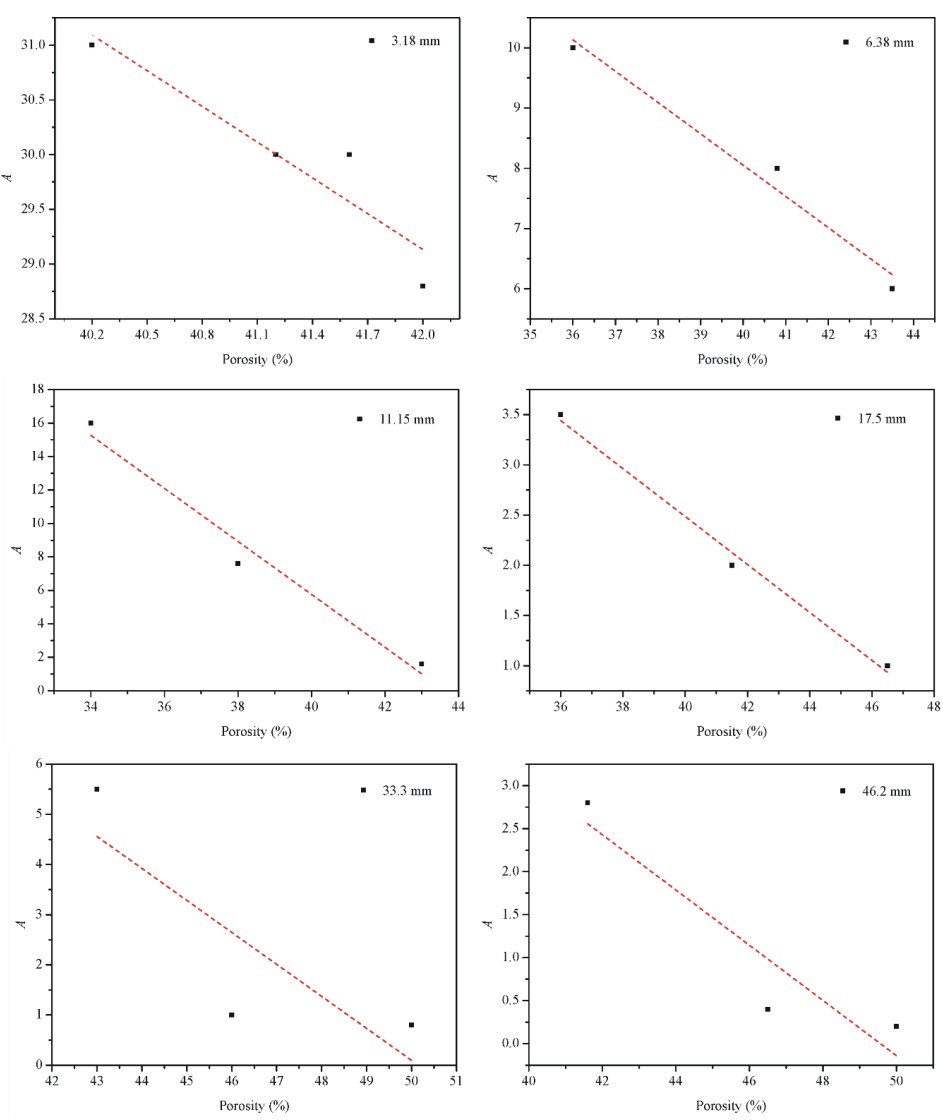

Fig. 17. Variation of $A$ with $n$ of six gravels with different particle sizes.

In view of this, we can use a macro parameter porosity ($n$) to reveal the effect of sorting

degree and arrangement on seepage coefficient. In order to verify the correctness of the above

analysis results, we selected the seepage experiment results of Niranjan (1973) for further

validation. Niranjan (1973) chose gravel of the same size but different porosity and carried

out seepage experiments. We selected the experimental results of six different particle sizes





with 3.18 mm, 6.38 mm, 11.15 mm, 17.5 mm, 33.3 mm and 46.2 mm from Niranjan (1973),
and drew the relationship between coefficient $A$ and $B$ and porosity respectively, as shown in
Fig. 17 and Fig. 18. We can see that the coefficients $A$ and $B$ of the six groups of experimental
data of Niranjan (1973) decrease with the increase of porosity, which is consistent with our
theoretical analysis of this investigation.

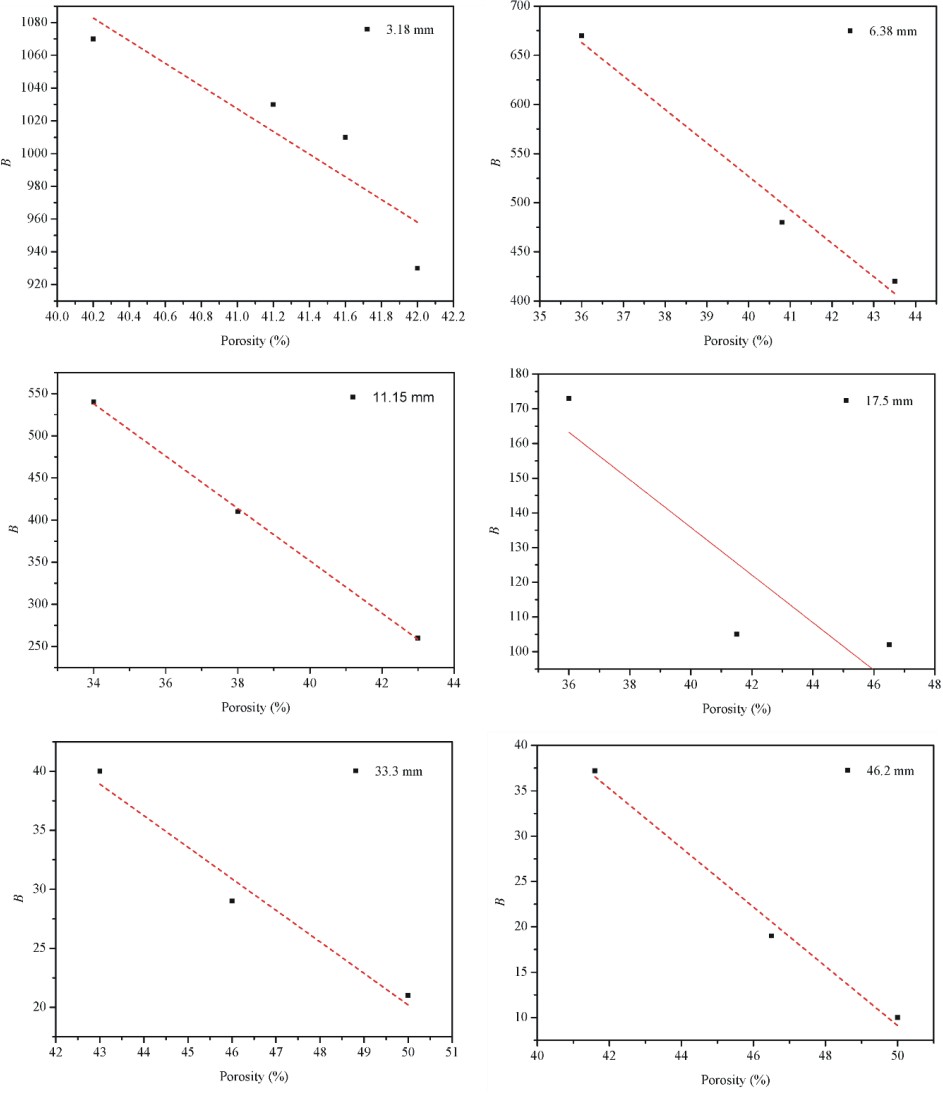


Fig. 18. Variation of $B$ with $n$ of six gravels with different particle sizes.



## 4. Summary and conclusions

This study presents experimental results of Forchheimer flow in four different permeable stones with different mesh sizes, including 24 mesh size (0.71 mm), 46 mesh size (0.36 mm), 60 mesh size (0.25 mm), 80 mesh size (0.18 mm). The effects of mean pore size and pore size distribution on the transition of flow regimes (from pre-Darcian to post-Darcian) are discussed. In addition, the mercury injection experiment is proposed to investigate the pore distribution of the permeable stones. In addition, the Forchheimer coefficients are specifically discussed. The main conclusions can be summarized as follows:

1) The relationships between specific discharge ($q$) and the "pseudo" hydraulic conductivity ($K$) (which is computed as a ratio of $q$ and hydraulic gradient, $J$) of permeable stones show that deviation from Darcian flow regime is clearly visible. In addition, the critical specific discharge corresponding to the transition of flow regimes (from pre-Darcian to post-Darcian) increases with the increase of mean particle size.

2) When the specific discharge is small, only a small fraction of the water flowing through the pores. The rest of the water adheres to the surface of the solid particles (immobile), partially blocking the flow pathways. As the specific discharge increases, more water becomes mobile and participates in flow. Hence, the "pseudo" hydraulic conductivity increases with the increase of specific discharge. When the specific discharge increases to the critical specific discharge ($q_c$), the "pseudo" hydraulic conductivity is maximized, and then it begins to decrease as the specific discharge continues to increase.

3) The mercury injection experiment results show that the mercury injection curve can be divided into three segments. The beginning and end segments of the mercury injection curve of the four permeable stones with different particle sizes are very gentle, while the main (or intermediate) mercury injection curve is steep, indicating that the pore size distribution falls within a narrow range, and the proportions of large pores and small pores are relatively small.





4) The porosity decreases as the mean particle size of permeable stone increases while the
mean pore diameter increases. And the porosity can reflect the influence of particle diameter,
sorting degree and arrangement mode of porous medium on seepage parameters. The larger
porosity leads to the smaller coefficients $A$ and $B$ under the condition of the same particle size.
5) The coefficient $A$ is linearly related to $1/d^2$ and the relationship between coefficient $A$ and
$1/d^2$ is given as $A = 0.0025\left(1/d^2\right) + 0.003$. The coefficient $B$ is not linearly related to $1/d$,
instead it is quadratic related to $1/d$ as $B = 1.14\text{E-06}\left(1/d\right)^2 - 1.26\text{E-06}\left(1/d\right)$. The particle
shape and arrangement of permeable stone have imposed great influences on the seepage
parameters.
**Notation**
$q$              The specific discharge, m/d.
$K$              The Hydraulic conductivity, m/d.
$J$              The dimensionless parameter defined as hydraulic gradient.
$A$              The Forchheimer equation coefficient (viscous force item), $\text{sm}^{-1}$.
$B$              The Forchheimer equation coefficient (Inertia force item), $\text{s}^2\text{m}^{-2}$.
$P_c$            The capillary force, $Pa$.
$P_{50}$         The corresponding pressure value when the saturation reaches 50%, $MPa$.
$P_A$, $P_B$, $P_C$    The pressure corresponding to different stages on mercury injection curve, $MPa$.
σ                The solid-liquid interfacial tension.
$\theta$         The wet angle between the liquid and the solid surface.
$r$              The radius of curvature in capillary, mm.
$d$              The particle size, mm.





$d_{50}$        The mean particle sizes (50% by weight), mm.
$R_m$        The mean pore diameter, $\mu m$.
$R_{50}$        The pore diameter corresponding to the median pressure $P_{50}$, $\mu m$.
$H$        The height of the peak of the mercury injection curve.
$x_c$        The abscissa corresponding to the peak of the curve (the pore size).
$w$        The standard variance.
$n$        The porosity.
$J_n$        The viscous force-related component.
$J_r$        The inertia force-related component.
**Authors contributions**
Zhongxia Li: Experiment, Writing original draft. Junwei Wan: Methodology,
Conceptualization. Tao Xiong: Data curation, Investigation, Experiment. Hongbin Zhan:
Methodology, Writing, Review & Editing. Linqing He: Experiment, Methodology. Kun
Huang: Funding acquisition, Investigation
**Competing interests**
The authors declare that they have no conflict of interest.
**Acknowledgements**

This study was supported by the National Natural Science Foundation of China (Grant

Nos. 41402204), the National Key Research and Development Program of China
(No. 2018YFC0604202) and the Fundamental Research Funds for National Universities,
China University of Geosciences (Wuhan). Thank Zhongzhi Shen of China University of
Geosciences for his great help in developing the experimental set up. And the authors want to



express their sincere appreciation of the constructive comments made by the three
anonymous reviewers and Associate Editor for improving the quality of the manuscript.

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
