# Peer review of "Experimental study of non-Darcian flow characteristics in permeable stones"

_Hydrology and Earth System Sciences, 2021_

## Author Comment (AC1)

College of Geosciences

Department of Geology & Geophysics, College Station, TX 77843-3115
Hongbin Zhan, Ph.D.
Professor of Geology and Geophysics, Professor of Water Management and Hydrological Sciences
Holder of Endowed Dudley J. Hughes Chair in Geology and Geophysics
Tel: (979) 862-7961, Fax: (979) 845-6162, Email: zhan@tamu.edu

[Figure]

January 19, 2022

Memorandum

To: Dr. Jorge Isidoro, Editor, Hydrology and Earth System Sciences

**Subject**: Revision of hess-2021-588

**Dear Editor:**

We have carefully revised our manuscript following all suggestions and comments of the reviewers. Thanks to the reviewers and editors, the manuscript has been improved substantially by addressing the constructive comments. The followings are responses to all the comments.

**Response to Reviewer #1:**
**Minor comments**

*I have completed the review of the manuscript entitled "Experimental study of non-Darcian flow characteristics in permeable stones" submitted to HESS for potential publication. In my experience, experimental research on non-Darcy flow is never out of data. In this manuscript, firstly, the seepage experiment of permeable stone provides experimental basis for non-Darcian seepage in relatively low permeability medium; then, pore distribution characteristics of various permeable stones are analyzed by mercury injection test with Gaussian distribution function; finally, the influences of particle size and porosity on Forchheimer equation coefficient are investigated and some interesting phenomena are found. This manuscript has potential to provide hints for non-Darcy studies, in terms of such as critical values of non-Darcy flow, influences of pore properties on non-Darcy flow in some specific views and enlightenment of some special phenomena. At this stage, I will recommend a minor revision since there are still some deficiencies that need to improve in this manuscript as follows:*

*1. Line 159: The basic information of permeable stone related to the manuscript topic, such as forming background and porous properties or generation, should be introduced firstly.*

**Reply:** Implemented. We have added the relevant application background and research status of permeable stone. Permeable stone is widely used in urban road design, sponge city construction and ecological effect research. And the most commonly used permeable base materials are large pore cement stabilized macadam, large diameter permeable asphalt mixture and so on. For permeable stone, there must be a certain connected pore space to maintain a certain permeability for transmitting water. However, the increase of pore space will lead to the decrease of pavement performance and mechanical strength. Therefore, many scholars have carried out a lot of research on controlling the proper pore space of permeable stone. Please see lines 179-187. In addition, we have outlined the preparation process of permeable stone. Please see lines 175-179.

*2. Line 226: As Fig. 4 indicates, the best-fitting yields Forchheimer numbers (F0=B/A=kβρν/μ) with orders of magnitudes to be about -4, but Zeng and Grigg (2006) suggested a critical F0 to be 0.11 to trigger high-velocity non-Darcian flow, which makes the flow in authors' seepage experiment looks like "super-weak non-Darcian type". If so, the authors should compare the best-fitting performances between Forchheimer equation and simple Darcy's law, to prove the necessity of existence of the inertial term of Bq2.*
*( Zeng, Z., & Grigg, R. (2006). A criterion for non-Darcian flow in porous media. Transport in Porous Media, 63(1), 57-69. https://doi.org/10.1007/s11242-005-2720-3)*

**Reply:** Implemented. In order to better compare with the actual groundwater flow, we converted the specific discharge to meters per day (m/d). Therefore, the best-fitting exercise yields Forchheimer numbers ($F_0= B/A$) is about $10^{-4}$ according to Fig. 4, which is fairly small. In addition, the critical Forchheimer numbers proposed by Zeng and Grigg (2006) and Javadi et al. (2014) are empirical, in fact, the transition between Darcy to non-Darcy is successional over a certain range of Forchheimer numbers. The non-Darcian flow criterion applicable to different pore media is established by conducting seepage resistance experiments in homogeneous and heterogeneous porous media in our previous study (Li et al., 2017; Li et al., 2019), which is consistent with the results of Zeng and Grigg (2006). Please see lines 227-236.

*3. Fig. 4: The results of best-fitting by Forchheimer equation have unconspicuous connection with the subsequent discussion of "pseudo" hydraulic conductivity and critical specific discharge.*

**Reply:** Implemented. Generally speaking, the $q$-$J$ and $q$-$K$ curves are the most common methods used to analyze flow regime when conducting seepage resistance experiments in porous media. However, the nonlinear characteristics of $q$-$J$ curve are not obvious due to the relatively small velocity range used in the experiments. The traditional hydraulic conductivity is the ratio of the specific discharge versus the hydraulic gradient ($q/J$), and it is a constant if Darcy's law is applicable, which is denoted as $K_D$ (Li et al., 2019). In fact, the ratio of $q/J$ is no longer a constant for the problems discussed in this study. In a word, the $q$-$K$ curve can be used to observe the transition of flow state more intuitively. Please see lines 236-243.

*4. Lines 299-300: The pressure ratio is a macroscopic parameter but the inhomogeneity is a relatively microscopic one, so the authors should prove the reasonability of PC/PB representing the inhomogeneity.*

**Reply:** Implemented. It is well known that for mercury injection experiments, as injection pressure increases, the injection saturation will gradually increase and eventually all the pores will be filled with mercury. As can be seen from Fig. 7, with the continuous injection of mercury, the pressure of permeable stones with different particle sizes varies with saturation, which is reflected in the different pressure $P_B$ and $P_C$ at different stages. However, the reason for the different pressure is the difference of pore size distribution in the permeable stones. Therefore, the pressure ratio of B and C ($P_C/P_B$) can be used as one of the criteria to characterize the heterogeneity of pore size in porous media. We have made relevant revision on this matter, please see lines 336-343.

*5. Equations (3-3) to (3-7) can be assembled into a single table for the purpose of more concise expression.*

**Reply:** Implemented. We have summarized a series of equation coefficients in Table 4 and revised the sentences. Please see lines 437-440 and Table 4.

*6. References should be provided for Equation (3-8). Eq. (3-8) cannot be derived from Eq. (1-2) alone.*

**Reply:** Implemented. We have added the relevant references. For specific derivation process, please refer to previous studies (Huang, 2012). Please see lines 466-467.

Reference cited in this reply:

Huang, K.: Exploration of the basic seepage equation in porous media, PhD dissertation, 2012.

Javadi, M., Sharifzadeh, M., Shahriar, K., and Mitani, Y.: Critical Reynolds number for nonlinear flow through roughˆ walled fractures: The role of shear processes, Water Resources Research, 50, 1789-1804, https://doi.org/10.1002/2013WR014610, 2014.

Li, Z., Wan, J., Huang, K., Chang, W., and He, Y.: Effects of particle diameter on flow characteristics in sand columns, International Journal of Heat & Mass Transfer, 104, 533-536, https://doi.org/10.1016/j.ijheatmasstransfer.2016.08.085, 2017.

Li, Z., Wan, J., Zhan, H., Cheng, X., Chang, W., and Huang, K.: Particle size distribution on Forchheimer flow and transition of flow regimes in porous media, Journal of Hydrology, 574, 1-11, https://doi.org/10.1016/j.jhydrol.2019.04.026, 2019.

Zeng, Z. and Grigg, R.: A criterion for non-Darcy flow in porous media, Transport in porous media, 63, 57-69, https://doi.org/10.1007/s11242-005-2720-3, 2006.

Please contact me if you have further questions.

Sincerely Yours,

Hongbin Zhan, Ph.D., P.G.

---

## Author Comment (AC2)

College of Geosciences

Department of Geology & Geophysics, College Station, TX 77843-3115
Hongbin Zhan, Ph.D.
Professor of Geology and Geophysics, Professor of Water Management and Hydrological Sciences
Holder of Endowed Dudley J. Hughes Chair in Geology and Geophysics
Tel: (979) 862-7961, Fax: (979) 845-6162, Email: zhan@tamu.edu

[Figure]

January 19, 2022

Memorandum

To: Dr. Jorge Isidoro, Editor, Hydrology and Earth System Sciences

**Subject**: Revision of hess-2021-588

**Dear Editor:**

We have carefully revised our manuscript following all suggestions and comments of the reviewers. Thanks to the reviewers and editors, the manuscript has been improved substantially by addressing the constructive comments. The followings are responses to all the comments.

**Response to Reviewer #2:**

*In this manuscript the authors perform experimental study of non-Darcian flow in four rock samples with different pore size distribution determined by mercury injection experiment. The authors determine the critical specific discharge and pre-Darcian flow regime using q-K curves for the rock samples. The Forchheimer coefficients are also determined form the experiments. It is an interesting work and, in my view, should be accepted after fixing the following minor issues:*

*1. The effective diameter (d10) of the pores is usually used to predict the permeability. As stated by Hazen (1892), the influences of the finer grain of the soil is more significant on pore space size and hydraulic conductivity comparing to that of coarser grain. In this work, however, mean grain size is used to draw a relation between hydraulic gradient and specific discharge. The authors should comment on this.*

**Reply:** Implemented! From the point of view of pore composition, the porous medium has been screened for the preparation of permeable stone, which can be regarded as homogeneous. The pore distribution is relatively concentrated over a narrow pore size range, and the proportion of large pores and small pores is very small. The average particle size can reflect the overall permeability of the porous media.

*2. Lines210-213: The particle size distribution of each sample should be given.*

**Reply:** Implemented! We have carried out the seepage experiments of permeable stones with four different mesh sizes in this study. The porous media used to prepare the permeable stone are carefully sieved and can be regarded as homogeneous. To facilitate the description, we can convert above four different mesh sizes of permeable stones into corresponding particle sizes. In other words, the four groups of permeable stones in this study are homogeneous porous media. Finally, we obtained the pore distribution of different permeable stones by carrying out mercury injection experiment, as shown in Fig. 8 to Fig. 11.

*3. Line 238: Izbash (1931) model is commonly used to simulate the pre-Darcy flow (Dejam et al., 2017). The authors should comment on this.*

**Reply:** Implemented! In fact, Izbash (1931) presented the equation as $q = M\left(\dfrac{dH}{dx}\right)^{m} = Mi^{m}$, where $M$ and $m$ are the coefficients determined by fluid flow and properties of porous media. When $m=1$, the Izbash equation reduces to Darcy law, when $m>1$, the Izbash equation corresponds to the pre-Darcy flow and when $m<1$, the Izbash equation refers to the post-Darcy flow (Dejam et al., 2017; Soni et al., 1978). Besides, Dejam et al. (2017) carried out a more detailed study on the issues related to the pre-Darcy and post-Darcy flows. And the influence of pre-Darcy flow on the pressure diffusion for homogenous porous media is studied in terms of the nonlinear exponent and the threshold pressure gradient. We have added the relevant information, please see lines 264-271.

*4. Lines 240-245: As another explanation: The pre-Darcy flow may also be due to an influence of the stream potential which generates the small countercurrent along pore walls in a direction opposite that of the main flow (Bear, 1972; Dejam et al., 2017).*

**Reply:** Implemented! In addition, another justification for the pre-Darcy behavior may be due to an effect of a stream potential which generates small countercurrents along pore walls in a direction opposite that of the main flow (Bear, 1972; Scheidegger, 2020). And Swartzendruber (1962a) stated that the surface forces arose in a solid-fluid interface due to strong negative charges on clay particle surfaces and the dipolar nature of water molecules caused a pressure gradient response to be nonlinear and led to the pre-Darcy flow (Swartzendruber, 1962b). We have supplemented the hypotheses of other scholars, please see lines 274-280.

Reference cited in this reply:
Bear, J.: Dynamics of Fluids in Porous Media, American Elsevier Pub. Co., New York, N.Y., and Amsterdam,1972.
Dejam, M., Hassanzadeh, H., and Chen, Z.: Pre‑Darcy flow in porous media, Water Resources Research, 53, 8187-8210, https://doi.org/10.1002/2017WR021257, 2017.
Izbash, S.: O Filtracii V Kropnozernstom Materiale, Leningrad, USSR, 1931.
Scheidegger, A. E.: The physics of flow through porous media, University of Toronto Press, https://doi.org/10.3138/9781487583750, 2020.
Soni, J., Islam, N., and Basak, P.: An experimental evaluation of non-Darcian flow in porous media, Journal of Hydrology, 38, 231-241, https://doi.org/10.1016/0022-1694(78)90070-7, 1978.
Swartzendruber, D.: Non‑Darcy flow behavior in liquid‑saturated porous media, Journal of Geophysical Research, 67, 5205-5213, https://doi.org/10.1029/JZ067i013p05205, 1962a.
Swartzendruber, D.: Modification of Darcy's law for the flow of water in soils, Soil Science, 93, 22-29, https://doi.org/10.1097/00010694-196201000-00005, 1962b.

Please contact me if you have further questions.

Sincerely Yours,

Hongbin Zhan, Ph.D., P.G.

---

## Editor Decision (ED1)

**Specific comments on HESS-2021-588**

- When using scientific notation use $10^n$ instead of E$n$). This can be noticed all along the manuscript, including in the figures and tables

- Please check if all the variables are listed in the notation list

- I suggest incorporating the author's reply to the first query of Reviewer #2 in the manuscript (in the Introduction or in section 2.2)

- Lines 24-25. Please rewrite this sentence and avoid repeating "mesh size"

- Line 24 and 49. I suggest using "…mercury injection technique…" instead of "… mercury injection experiment…"

- I strongly suggest not starting sentences with "And…" (*e.g.*, Lines 181, 270, 278, 293, 312, 357, 428, 441, 528, 589)

- Line 129. Please use "five" instead of "5".

- Line 178. An "a" is missing between "permeable" and "stone" (unless the authors are referring the preparation of a sample in general; if so, "stones" should be used).

- Lines 180-183. Please revise these sentences. Saying that "Permeable stone is widely used in […] ecological effect research" does not sound good

- Lines 183-185. This sentence is tremendously ambiguous, as "certain connected pore space" and "certain permeability" does not define what can, or cannot, be considered a "permeable stone"

- Line 188. I suggest starting a new paragraph with "We have carried out…"

- Lines 188-192. Please rewrite this sentence and avoid repeating "mesh size"

- Line 228 and 230. Eliminate "(m/d)" or use (m d$^{-1}$) instead

- Lines 219-223 is repeated (see Lines 188-194)

- Line 232. I suggest starting a new sentence with "In fact, the transition…"

- Line 280. Please add a comma before "and the dipolar…"

- Line 323 (Figure 6). Please capitalize "Saturation" (Y-axis)

- Line 411 (Table 2). Please capitalize "Particle size" (header)

- Line 431 (Table 3). It is essential to repeat the "mean particle size" column also presented in Table 2?

- Lines 438-440. Please rewrite this sentence, which is too long and confusing

- Table 4. Use (s m$^{-1}$) and (s$^2$ m$^{-2}$)

- Lines 457-459. Please check this sentence

- Line 461. RMSE is well-known. It is unnecessary to explain it or present the equation

- Lines 473-476. Please separate into two sentences. The second sentence starts with the reference (Huang et al.)

- Line 525. Please check this figure's caption as it sounds awkward

- Line 503-505. Please check this sentence

- Line 569. Please avoid repeating "In addition"

- Figures 15 and 16. These figures can be reduced without losing information

- Line 600. "The hydraulic gradient" is enough.

- Some units are missing in the notation list.

---

## Author Response (AR2)

College of Geosciences

Department of Geology & Geophysics, College Station, TX 77843-3115
Hongbin Zhan, Ph.D.
Professor of Geology and Geophysics, Professor of Water Management and Hydrological Sciences
Holder of Endowed Dudley J. Hughes Chair in Geology and Geophysics
Tel: (979) 862-7961, Fax: (979) 845-6162, Email: zhan@tamu.edu

[Figure]

April 13, 2022

Memorandum

To: Dr. Jorge Isidoro, Editor, Hydrology and Earth System Sciences

**Subject**: Revision of hess-2021-588

**Dear Editor:**

We have carefully revised our manuscript following all suggestions and comments, checked the English language and improved the writing of the manuscript by the help of a native English speaker. The manuscript has been improved substantially by addressing the constructive comments. The followings are responses to the comments, and the line number is according to the revised manuscript.

**Response to Editor:**
**Minor revision**

*Dear Authors.*
*As said before, this theme fits well in the contents of this special issue, and the manuscript presents an interesting experimental study on non-Darcian flow characteristics in permeable stones.*
*I've read the revised version of the manuscript with attention. In my opinion, the manuscript is interesting, and the minor issues detected by the reviewers were generally well addressed. However, the quality of English is clearly below the minimum required for an international journal such as HESS. Some sentences are repetitive, awkward, confusing, too long, or simply not elegant. Moreover, there are too many inconsistencies in the way things are presented (e.g., the meaning of variables shown in equations; the font size in the figures). Apart from this, the authors are not following many of the guidelines provided by HESS, an issue that also needs to be addressed. Regarding the latter, I urge the authors to carefully read and follow HESS' guidelines for authors before making further changes to the manuscript.*
*I invite the authors to improve the syntax and grammar of the text. Many phrases must be rewritten for the sake of ease of reading and comprehensibility. Please take also into attention the specific comments listed in the attached file. A paper should be joyful to read, and this manuscript still does need work to attain that level.*
*Non-public comments to the Author:*
*I suggest the authors find a professional editing service, or a native English speaker within this field of knowledge, to assist in producing a better manuscript. Honestly, I liked the science in the paper, but it is poorly presented.*

*1. When using scientific notation use $10^n$ instead of En). This can be noticed all along the manuscript, including in the figures and tables.*

**Reply:** Implemented. We have carefully revised the presentation of the scientific notation for the manuscript, including all the figures and tables.

108 Halbouty
3115 TAMU
College Station, TX 77843-3115

Tel. 979.845-2451 Fax 979.845-61627
Geoweb.tamu.edu

*2. Please check if all the variables are listed in the notation list.*

**Reply:** Implemented. We have carefully checked the whole manuscript and modified the notation list, the relevant variables and units were added. Please see page 36 and lines 621-625.

*3. I suggest incorporating the author's reply to the first query of Reviewer #2 in the manuscript (in the Introduction or in section 2.2)*

**Reply:** Implemented. We have added the relevant replies to the section 2.2 of the manuscript. Please see page 9 and lines 207-209.

*4. Lines 24-25. Please rewrite this sentence and avoid repeating "mesh size"*

**Reply:** Implemented. We have rewritten the sentence. Please see page 2 and lines 27-29.

*5. Line 24 and 49. I suggest using "…mercury injection technique…" instead of "… mercury injection experiment…"*

**Reply:** Implemented. We have adjusted some of the expressions. Please see page 2, 3, 15, 35 and lines 38, 54, 322, 584.

*6. I strongly suggest not starting sentences with "And…" (e.g., Lines 181, 270, 278, 293, 312, 357, 428, 441, 528, 589)*

**Reply:** Implemented. We have corrected these sentences and the whole manuscript was checked.

*7. Line 129. Please use "five" instead of "5".*

**Reply:** Implemented. We have corrected it. Please see page 6 and line 137.

*8. Line 178. An "a" is missing between "permeable" and "stone" (unless the authors are referring the preparation of a sample in general; if so, "stones" should be used).*

**Reply:** Implemented. We have made corrections referring to the preparation of general samples. Please see page 9 and line 184.

*9. Lines 180-183. Please revise these sentences. Saying that "Permeable stone is widely used in […] ecological effect research" does not sound good*

**Reply:** Implemented. We have rewritten these sentences. Please see page 9 and lines 186-190.

*10. Lines 183-185. This sentence is tremendously ambiguous, as "certain connected pore space" and "certain permeability" does not define what can, or cannot, be considered a "permeable stone"*

**Reply:** Implemented. We have corrected this part of the expression. Please see page 9 and line 194.

*11. Line 188. I suggest starting a new paragraph with "We have carried out…"*

**Reply:** Implemented. We have made corrections. Please see page 9 and line 200.

*12. Lines 188-192. Please rewrite this sentence and avoid repeating "mesh size"*

**Reply:** Implemented. We have rewritten these sentences. Please see page 9 and lines 200-202.

*13. Line 228 and 230. Eliminate "(m/d)" or use (m d-1) instead*

**Reply:** Implemented. We have corrected it. Please see page 11 and lines 243-245.

*14. Lines 219-223 is repeated (see Lines 188-194)*

**Reply:** Implemented. We have modified these two parts. Please see page 11 and lines 234-239.

*15. Line 232. I suggest starting a new sentence with "In fact, the transition…"*

**Reply:** Implemented. We have corrected it. Please see page 12 and line 247.

*16. Line 280. Please add a comma before "and the dipolar…"*

**Reply:** Implemented. We have added a comma. Please see page 14 and line 296.

*17. Line 323 (Figure 6). Please capitalize "Saturation" (Y-axis)*

**Reply:** Implemented. We have replaced the Figure 6.

*18. Line 411 (Table 2). Please capitalize "Particle size" (header)*

**Reply:** Implemented. We have corrected it. Please see Table 2.

*19. Line 431 (Table 3). It is essential to repeat the "mean particle size" column also presented in Table 2?*

**Reply:** Implemented. We have corrected it. Please see Table 3.

*20. Lines 438-440. Please rewrite this sentence, which is too long and confusing*

**Reply:** Implemented. We have rewritten these sentences. Please see page 25 and lines 456-459.

*21. Table 4. Use (s m-1) and (s2 m-2)*

**Reply:** Implemented. We have corrected it. Please see Table 4.

*22. Lines 457-459. Please check this sentence*

**Reply:** Implemented. We have rewritten these sentences. Please see page 25 and lines 462-464.

*23. Line 461. RMSE is well-known. It is unnecessary to explain it or present the equation*

**Reply:** Implemented. We have simplified this part. Please see page 26 and lines 468-471.

*24. Lines 473-476. Please separate into two sentences. The second sentence starts with the reference (Huang et al.)*

**Reply:** Implemented. We have rewritten these sentences. Please see page 26 and lines 481-483.

*25. Line 525. Please check this figure's caption as it sounds awkward*

**Reply:** Implemented. We have corrected the caption. Please see page 31 and lines 540-544.

*26. Line 503-505. Please check this sentence*

**Reply:** Implemented. We have rewritten these sentences. Please see page 29 and lines 514-517.

*27. Line 569. Please avoid repeating "In addition"*

**Reply:** Implemented. We have corrected it. Please see page 35 and line 585.

*28. Figures 15 and 16. These figures can be reduced without losing information*

**Reply:** Implemented. We have combined the Figure 15 and Figure 16. Please see Figure 15.

*29. Line 600. "The hydraulic gradient" is enough.*

**Reply:** Implemented. We have corrected it. Please see page 36 and line 618.

*30. Some units are missing in the notation list.*

**Reply:** Implemented. We have added the relevant information. Please see pages 36, 37 and lines 616, 617, 629.

Please contact me if you have further questions.
Sincerely Yours,
Hongbin Zhan, Ph.D., P.G.